# Environmental Remediation of Desalination Plant Outfall Brine Discharge from Heavy Metals and Salinity Using Halloysite Nanoclay

**Naif S. Aljohani** [1], **Radwan K. Al-Farawati** [2] , **Ibrahim I. Shabbaj** [1], **Bandar A. Al-Mur** [1], **Yasar N. Kavil** [2] and **Mohamed Abdel Salam** [1,3,*]

[1] Department of Environmental Sciences, Faculty of Meteorology, Environment and Arid Land Agriculture, King Abdulaziz University, P.O. Box 80208, Jeddah 21589, Saudi Arabia; nssh56@gmail.com (N.S.A.); Ishabbj@kau.edu.sa (I.I.S.); balmur@kau.edu.sa (B.A.A.-M.)

[2] Marine Chemistry Department, Faculty of Marine Sciences, King Abdulaziz University, P.O. Box 80207, Jeddah 21589, Saudi Arabia; rfarawati@kau.edu.sa (R.K.A.-F.); fidayas@gmail.com (Y.N.K.)

[3] Department of Chemistry, Faculty of Science, King Abdulaziz University, P.O. Box 80200, Jeddah 21589, Saudi Arabia

[*] Correspondence: masalam16@hotmail.com; Tel.: +966-541886660; Fax: +966-2-6952292

**Abstract:** Halloysite (HS) nanoclay was used for the environmental treatment of desalination brine water discharge via the adsorptive removal of selected heavy metals ions; zinc, iron, nickel, and copper, as well as salinity. Different techniques were used for the characterization of the HS nanoclay and it was found that HS nanoclay exists as transparent hollow nanotubes with high surface area. The study showed that most of the heavy metal ions could be removed successfully using the HS nanoclay in a few minutes, at normal conditions. The adsorptive removal of zinc, iron, nickel, and copper, as well as salinity on HS nanoclay was explored kinetically. It was concluded that the pseudo-second-order kinetic model was able to describe the remediation process. In addition, it was found that most of the heavy metals and salinity were removed from the desalination plant outfall brine discharge and the final concentrations were lower than those in the control and standard samples.

**Keywords:** brine; kinetics; heavy metals; nanoclay; remediation; salinity

## 1. Introduction

Freshwater resource scarcity is a major problem for many countries worldwide. The United Nations (UN) report (Environment Program) has demonstrated that one-third of the world's population has insufficient freshwater [1]. At the same time, 97.5 percent of the Earth's water is found in the oceans and seas [2,3]. The salinity range in seawater is 35,000–45,000 ppm [4–7], but the permissible salinity level in drinking water is 500 ppm and up to 1000 ppm for special cases [8]. This shows the necessity of desalination plants in the present era. The desalination plants discharge effluents (brine) are characterized by the presence of high salinity and high heavy metal contents which usually lead to contamination of the marine environment in the vicinity of desalination plants [9]. The current brine remediation process is mostly based on the discharge of the brine into the water bodies, i.e., the sea, and oceans, or to the land. This creates many environmental problems including the disturbance of the aquatic system balance and damage to the fauna and flora around the desalination plants [10]. In addition, the presence of heavy metal ions, such as zinc, iron, nickel, and copper in the aquatic environment, even at low concentrations, might cause a serious issue due to their toxicity and carcinogenicity to living organisms. Although, elements like copper and zinc are essential for living organisms, but similar to all heavy metals, they are harmful at excess concentrations, as excess of zinc ions could cause vomiting, nausea, and hematemesis, whereas excess of copper ions could cause Wilson's disease [11], nickel ions could cause cancer [12], and a long-term exposure to iron ions may be preferentially toxic to cells with high mitochondrial activity [13]. In addition, it is

well known that heavy metals do not biologically degrade similar to the organic pollutants and could accumulate in living organism's tissues. Thus, their existence in industrial effluents and drinking water is a major public health issue due to their introduction to the food chain by untreated waste discharge into water. Accordingly, more rigorous conditions and standards for the removal and elimination of heavy metals and high salinity from the brine in order to discharge them into the environment are mandatory. Although many processes are commonly applied for the treatment of the brine discharge salinity, such as the zero liquid discharge [14], membrane-based technologies including membrane coagulation [15], forward osmosis [16], reverse osmosis and high-pressure reverse osmosis [17], osmotically assisted reverse osmosis [18], membrane crystallization [19], different electrodialysis [20–22], as well as thermal-based technologies [23,24], the processes which remove heavy metals from brine are scarce in the literature [25,26]. On the other hand, various methods were used for the treatment of wastewater containing heavy metals, but remediation by adsorption has many advantages, such as the ease of the procedure, high performance, cost effectiveness, and the ability to regenerate/reuse/recycle both adsorbent and adsorbate [27–30]. Accordingly, the search for new types of adsorbents is the main focus of researchers worldwide. Nanoclays are a modern class of adsorbents composed of nano-dimensional coated mineral silicates that are inexpensive and non-hazardous, and with high surface reactivity and durability. Recently, nanoclays were used as potential adsorbents for heavy metal removal from different aquatic environments [31–34].

The present research work's objective is to explore, for the first time, the potential application of Halloysite nanoclay (HS nanoclay) for the treatment of an outfall brine discharge sample of the Yanbu Desalination Plant (YDP), Saudi Arabia, from both heavy metals; zinc, iron, nickel, and copper, and salinity to permissible levels. The morphological characterization of the HS nanoclay was performed by transmittance electron microscopy, X-ray diffraction, as well as surface area analysis. The HS nanoclay was then applied for the elimination/removal of zinc, iron, nickel, and copper ions, and salinity, from a brine discharge sample. The influence of different experimental conditions that may influence the remediation process was optimized, and the kinetics and thermodynamics of the removal were explored for the better understanding of the remediation process.

## 2. Materials and Methods

### 2.1. Materials

Two samples were collected from the site. The desalination plant outfall brine discharge sample was collected from the desalination plant at Yanbu city by the Red Sea coast of Saudi Arabia (latitude 23.916598, longitude 38.303265), as it is presented in Figure 1, and the control sample, which represents the uptake of the desalination plant. A 0.45 μm acid-clean Millipore filter paper was used to filter the outfall brine discharge sample and the control sample, and then stored in darkness at 5 °C using acid-clean Teflon® bottles.

HS nanoclay was obtained from Sigma-Aldrich Canada (Oakville, ON, Canada) (685445 Aldrich), and all other chemicals were analytical grade and obtained from Sigma-Aldrich USA (St. Louis, MO, USA). In addition, all solutions were prepared using Milli-Q water.

### 2.2. Characterization

A JEOL JEM-1230 transmission electron microscope (TEM) was used to examine the morphology of the HS nanoclay. X-ray diffraction (XRD) patterns were obtained on a Philips X-pert pro diffractometer. The specific surface area was measured with the A NOVA 3200e automatic gas sorption system (Quantachrome, Boynton Beach, FL, USA).

### 2.3. Removal Experiments

A certain amount of the HS nanoclay was added to 60 mL of the outfall brine discharge sample, then stirred for a specific period at ambient temperature, and after a definite time, the solution was filtered, and the concentrations of the residual heavy metal ions were

determined using the stripping voltammetry technique. The removal efficiency ($R\%$) and capacity ($q_t$) of the heavy metal ions by the HS nanoclay was estimated using the following equations:

$$R\% = 100 \times \frac{C_0 - C_t}{C_0} \tag{1}$$

$$q_t = (C_0 - C_t) \times \frac{V}{m} \tag{2}$$

where $C_0$ and $C_t$ are the concentrations of the desired heavy metal ions in the outfall brine discharge sample (mg/L), before ($C_0$) and after ($C_t$) the treatment, $m$ is the mass of the HS nanoclay (g), and $V$ is the solution volume (L). All the experiments were conducted in triplicate, and the stated values were the average value with less than 5% error.

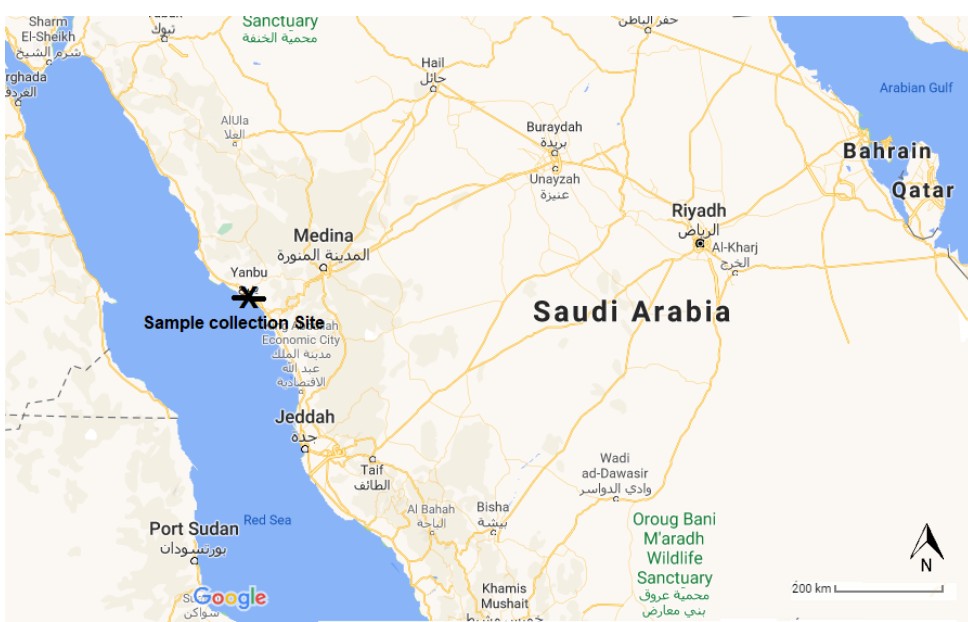

**Figure 1.** A map presenting the collection site by the Red Sea at the city of Yanbu, Saudi Arabia.

*2.4. Heavy Metal Ions Measurements*

The zinc, iron, nickel, and copper ion concentrations were measured voltammetrically using a Metrohm, 797 VA. The working electrode was a hanging mercury drop electrode (HMDE). The reference electrode was a double-junction electrode. An Ag/AgCl/3M KCl reference cartridge (Metrohm, Switzerland) was separated by a frit from a salt bridge filled with 3 M KCl. The bridge was freshly filled with a KCl solution at the beginning of the experiments. The counter electrode was a glassy carbon rod. The voltammetric cell was made of glass and solutions in the cell were stirred using a rotating Polytetrafluoroethylene (PTFE) rod. The metal ion determination was evaluated by adsorptive cathodic stripping voltammetry (AdCSV). The reagents used for the measurement of Cu was Salicylaldoxime (SA) and boric acid [35]. Dimethylglyoxime (DMG) and boric acid were used for the measurement of Ni, and Ammonium pyrrolidine dithiocarabamate (APDC) and boric acid were used for the measurement of Zn [36,37]. In the case of Fe, 2,4-dihydroxynaphthalene (DHN), HEPES (4-(2-Hyxdroyethyl) piperazine-1-ethanesulfonic acid), and potassium bromate were used [38]. The accuracy of the method was tested by analyzing near shore seawater reference materials for trace metals (CASS-4). Our results were within 15% of the certified values.

## 3. Results and Discussion

*3.1. HS Nanoclay Characterization*

The TEM images shown in Figure 2 revealed that the HS nanoclay was characterized with the formation of hollow tubes with an average diameter of 60 nm and an average

length of 10 microns. Figure 3 illustrates that the XRD pattern of the HS nanoclay characteristic peaks were perfectly indexed to the standard JCPDS file no.29-1487. Figure 4 shows the N2 adsorption/desorption isotherms for the HS nanoclay was a type III isotherm and the specific surface (BET) was 72.8 $m^2$ $g^{-1}$. Figure 5 shows the FTIR spectrum of the HS nanoclay, and the characteristic peaks of the HS nanoclay appear at 3696 $cm^{-1}$, 3629 $cm^{-1}$, and 1654 $cm^{-1}$ and are assigned to the O–H stretching of the inner-surface hydroxyl groups, the O–H stretching of inner hydroxyl groups, and deformation of water, respectively. In addition, the inner-surface OH groups are connected to the Al-centered octahedral sheets and form hydrogen bonds with the oxygen sheet in the next double layer [39].

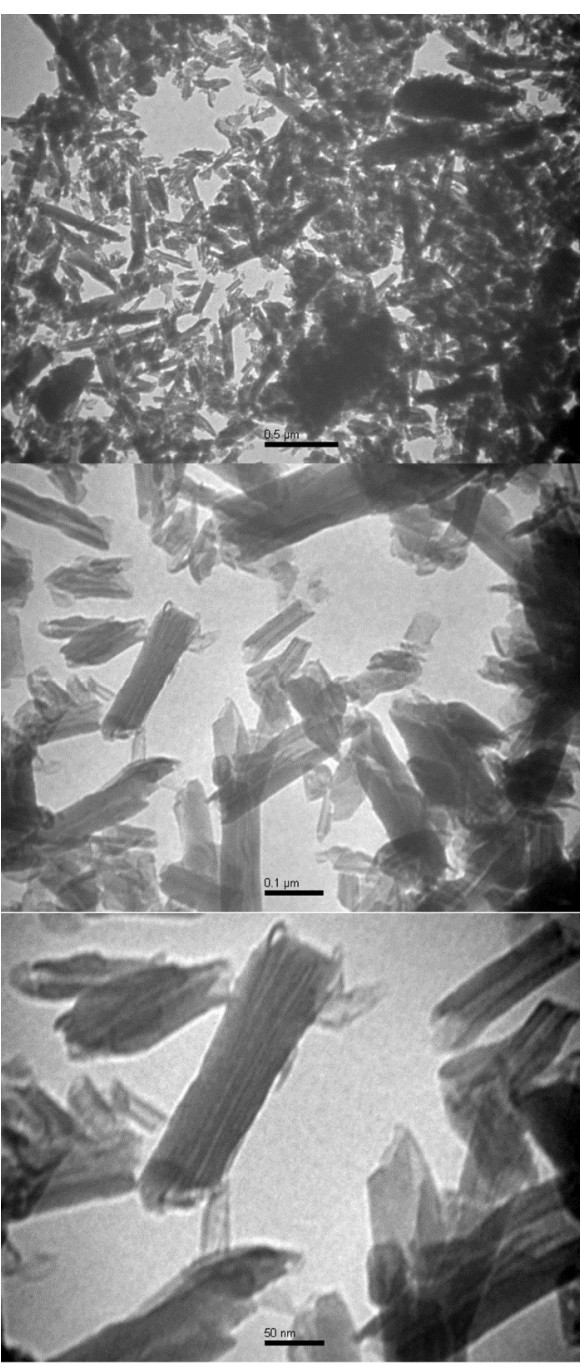

**Figure 2.** Transmission electron microscope (TEM) images of Halloysite nanoclay at different magnification powers.

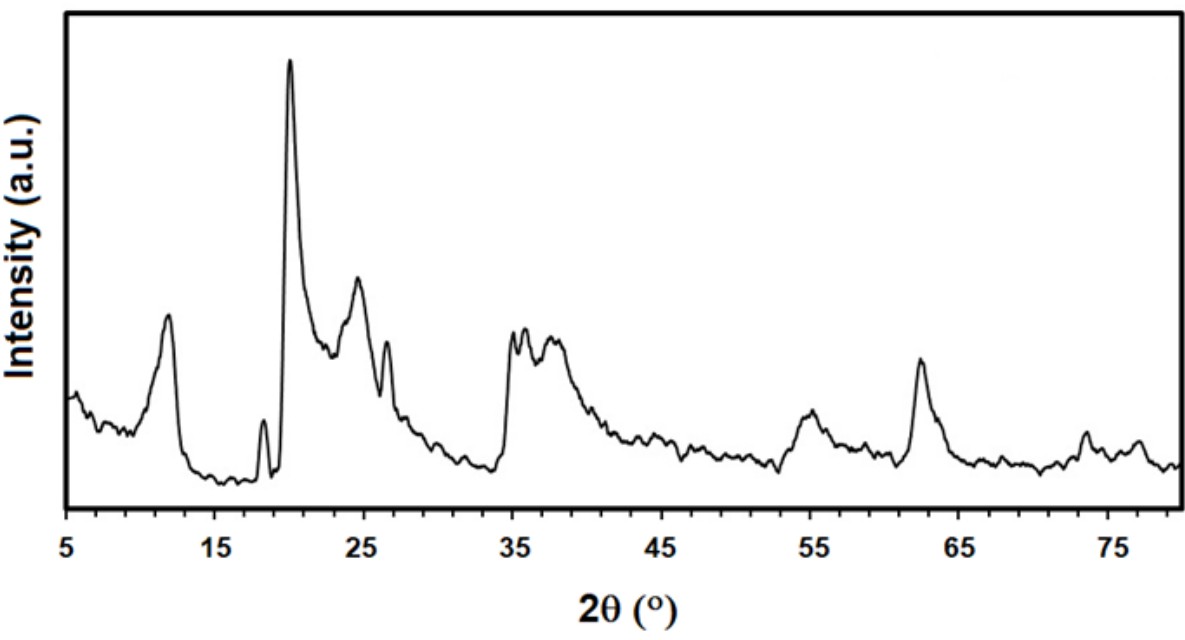

**Figure 3.** X-ray diffraction (XRD) pattern of Halloysite nanoclay.

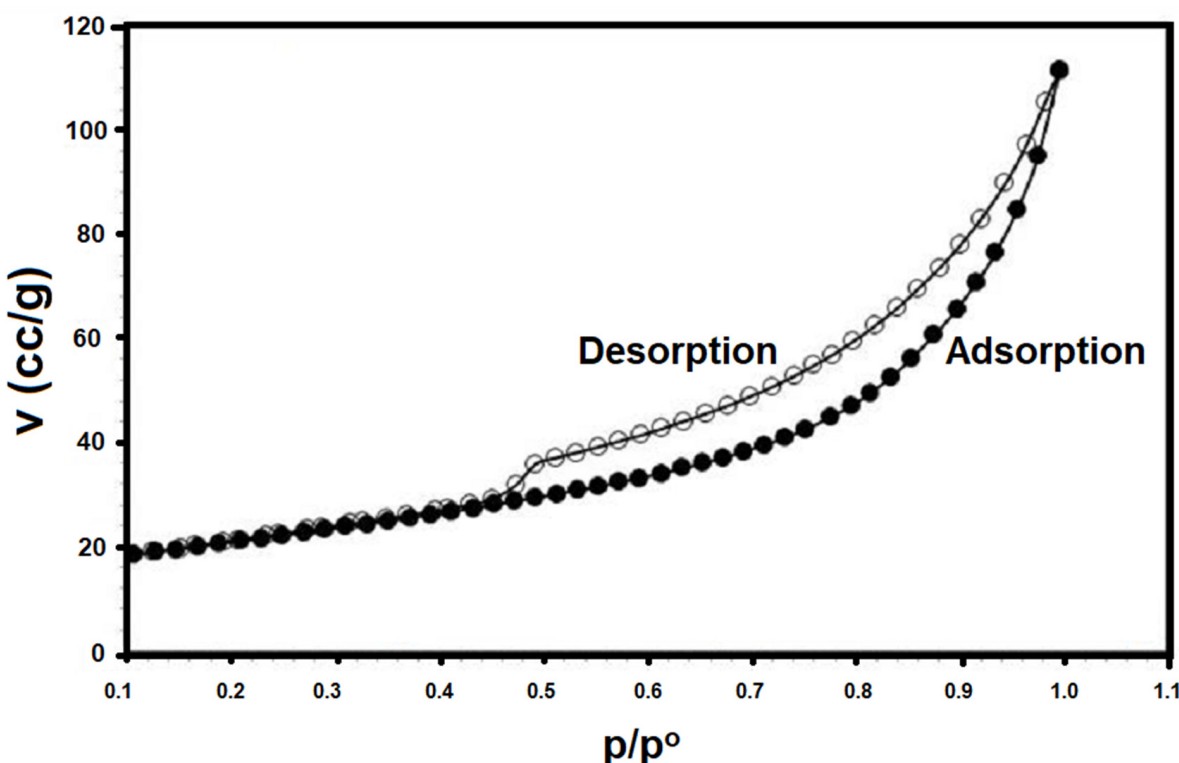

**Figure 4.** BET surface area of halloysite nanoclay at 77 K.

### 3.2. Outfall Brine Discharge and Control Samples Characterization

The concentrations of the heavy metals of the brine discharge sample were measured and were found to be 103.8 nM (6.8 ng/mL, ppb), 15.3 nM (0.85 ng/mL), 39.4 nM (2.3 ng/mL), and 117.1 nM (7.4 ng/mL), for the zinc, iron, nickel, and copper ions, respectively, whereas the salinity concentration was 48.0 g/L (ppt). On the other hand, the concentrations of the heavy metals of the control sample were 45.2 nM (3.0 ng/mL), 3.67 nM (0.20 ng/mL), 13.9 nM (0.81 ng/mL), and 28.6 nM (1.8 ng/mL), for the zinc, iron, nickel, and copper ions, respectively, with a salinity concentration of 42.2 g/L (ppt).

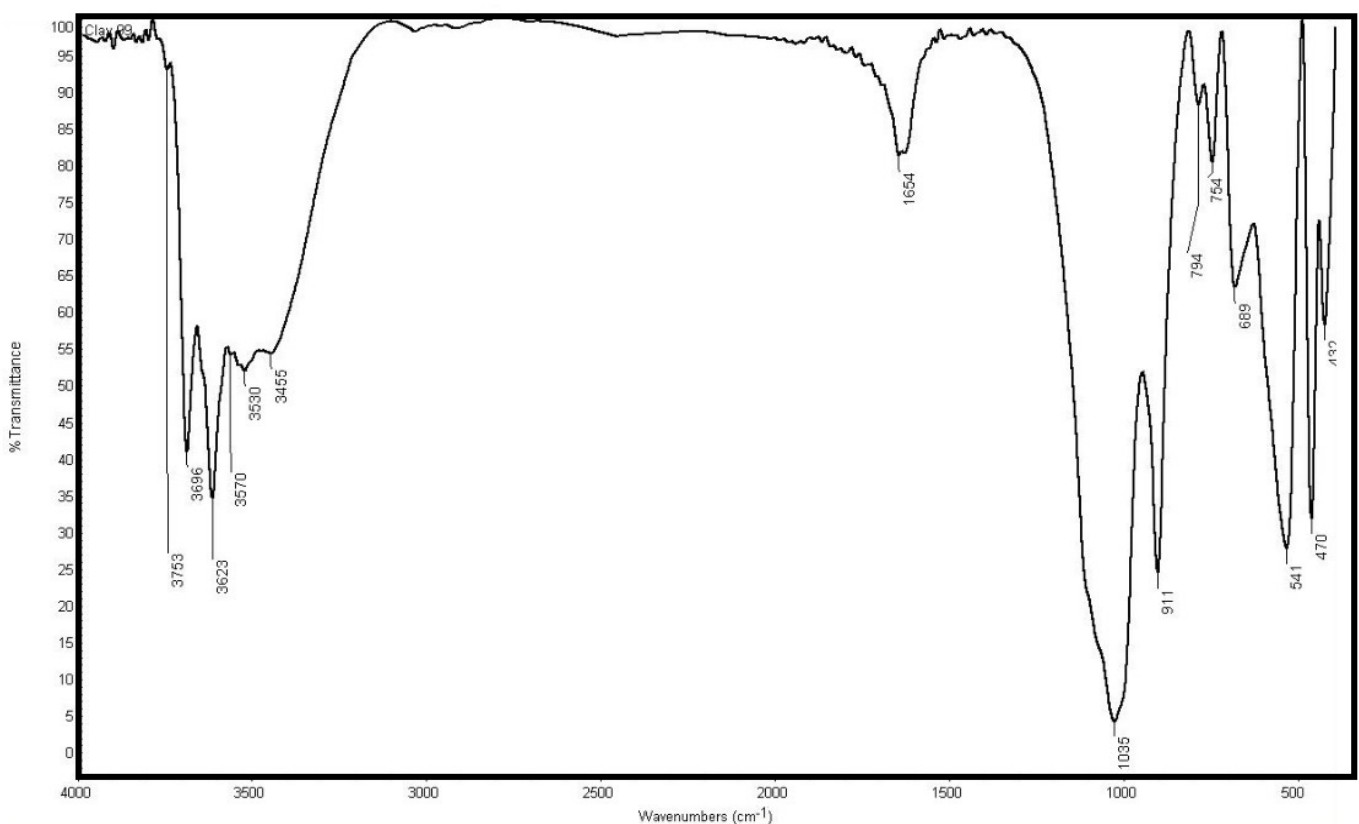

**Figure 5.** FTIR spectrum of HS nanoclay.

### 3.3. Adsorption Study

Environmental remediation and treatment of the polluted aquatic environment such as brine is generally governed by different operational and experimental factors that greatly affect the remediation process. Therefore, the effect of various experimental factors which may affect the removal of Zn, Fe, Ni, and Cu ions from the desalination outfall brine discharge sample by the HS nanoclay were explored and optimized. First, the effect of the HS nanoclay mass was explored and the results are explained in Figure 6. It illustrates that increasing the HS nanoclay mass till 150 mg greatly enhanced the removal efficiency, which reached 86.7%, 86.0%, 84.2%, and 86.7%, for the Zn, Fe, Ni and Cu ions, respectively, and the remaining/unremoved concentration of the Zn, Fe, Ni, and Cu ions were 13.8 nM, 2.14 nM, 6.24 nM, and 15.6 nM, respectively. This is significantly lower than the concentrations in the control sample. In addition, the salinity was reduced from 48.0 g/L to 41.1 g/L which is lower than the salinity in the control sample; 42.2 g/L. On the other hand, the application of a HS nanoclay mass of more than 150 mg, was accompanied by a decrease in the removal of Zn, Fe, Ni, and Cu ions due to the difficulty for homogenous mixing, as the amount of HS nanoclay was not well mixed with the outfall brine discharge sample.

The variation of the adsorption/removal of Zn, Fe, Ni, Cu ions from the outfall brine discharge with time by the HS nanoclay was explored, and the results are presented in Figure 7. There was a clear enhancement of the percent removal for the four metals ions with increasing adsorption time within the first 60 min and the removal reached 78.35% for Zn, 81.97% for Fe, 79.75% for Ni, and 77.56% for Cu. Increasing the adsorption time was associated with an insignificant increase in the removal: 86.12% for Zn, 86.59% for Fe, 84.08% for Ni, and 86.37% for Cu. Accordingly, the rest of the experiments were performed using 90 min as the adsorption time to assure the equilibrium.

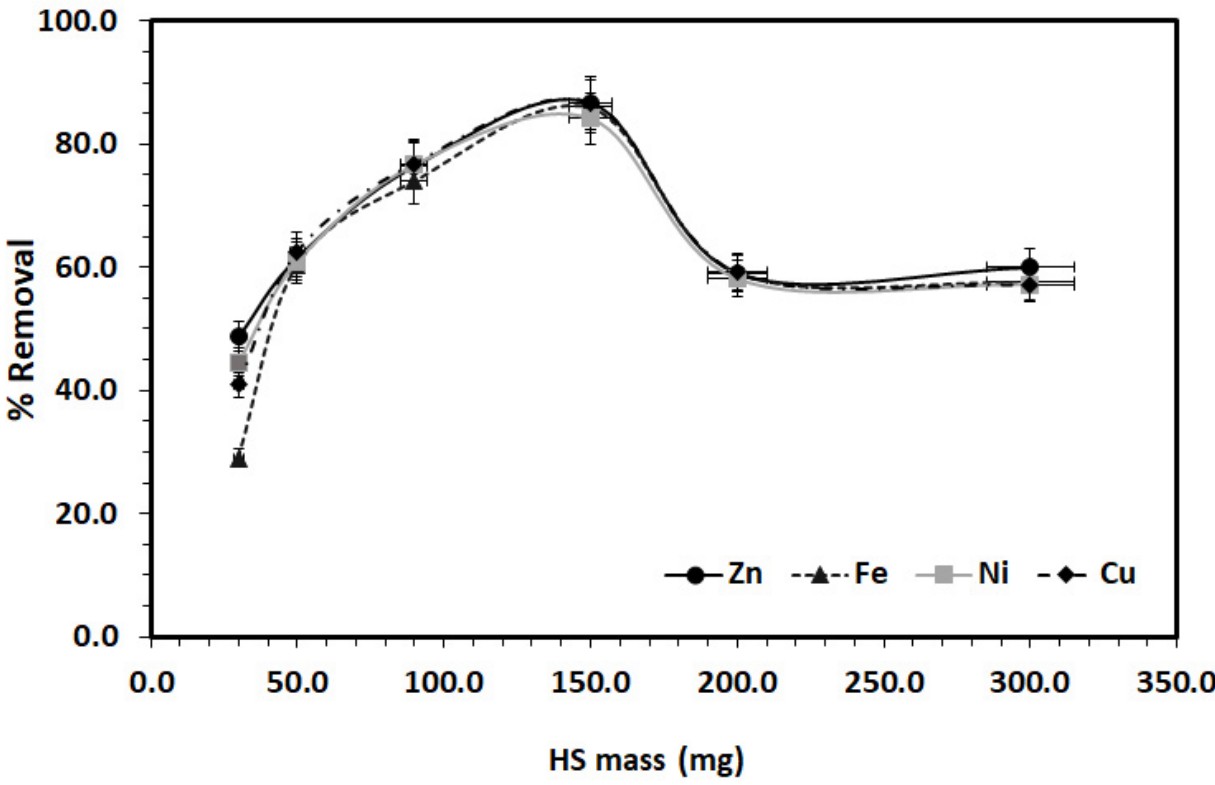

**Figure 6.** Effect of HS nanoclay mass on the removal of metal ions (Zn, Fe, Ni, Cu) from a desalination plant outfall brine discharge. (Experimental conditions: 60.0 mL solution, pH 8.0, 90 min adsorption time, and 25 °C temperature).

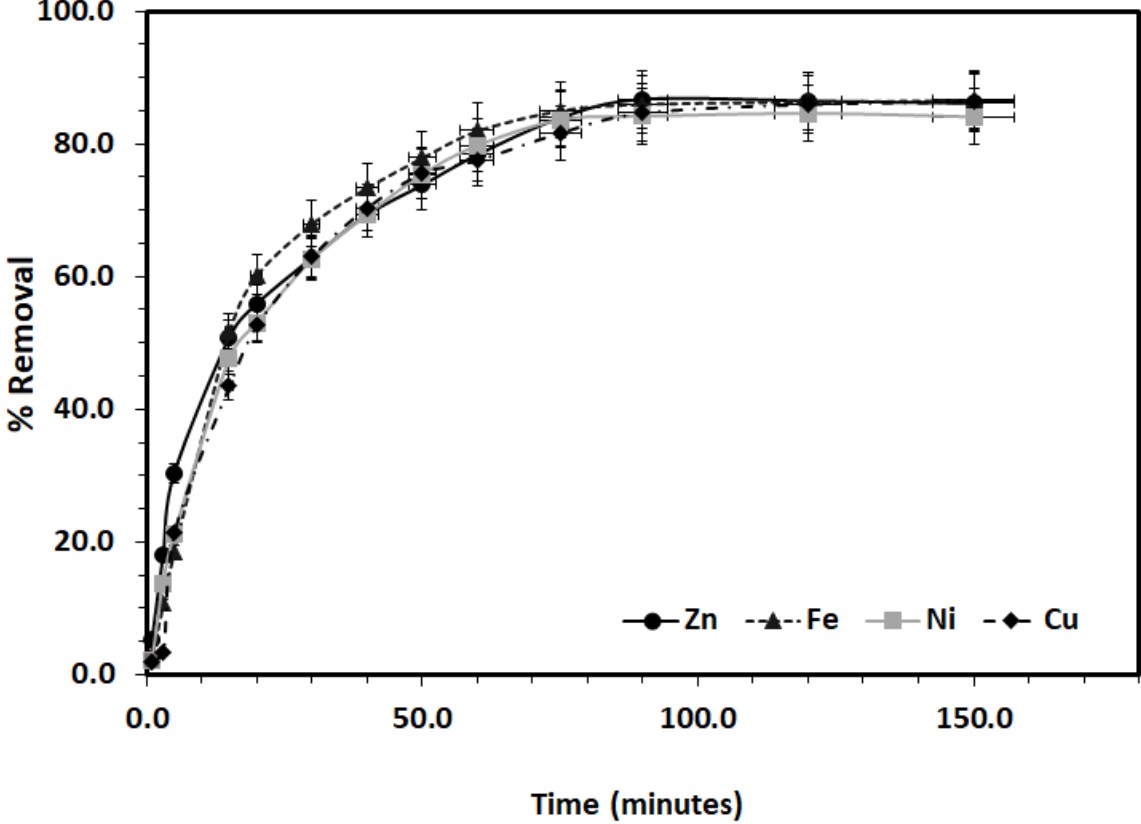

**Figure 7.** Effect of contact time on the removal of metal ions (Zn, Fe, Ni, Cu) adsorption from a desalination plant outfall brine discharge by HS nanoclay. (Experimental conditions: 60.0 mL solution, pH 8.0, 150 mg dosage, and 298 K).

The effect of the solution pH on the adsorption/removal of Zn, Fe, Ni, Cu ions from the outfall brine discharge using HS nanoclay was explored at pH values ranging from 2.0 to 10.0, as it is illustrated in Figure 8. In general, at pH values of 2.0 to 4.0, the hydronium ions ($H_3O^+$) concentration is very high compared with metal ions and accordingly outcompete with the Zn, Fe, Ni, Cu ions for the active binding sites at the HS nanoclay surface, which in turn causes low metal ions removal. In addition, at high pH values of 6 to 8, the percent removal was significantly enhanced for all the metal ions, and reached the highest percentage at pH values of 8.0; 86.23% for Zn, 85.73% for Fe, 85.96% for Ni, and 86.80% for Cu. This is may be due to the change of the HS nanoclay surface charge to negative, which enhanced the adsorption of the positive cations metal ions via electrostatic attraction [26]. Increasing the solution pH to 10 was associated with a significant reduction of the adsorption to 34.91% for Zn, 22.9% for Fe, 29.80% for Ni, and 27.42% for Cu, which may be due to the competition of the hydroxide ions with the selected metal for the adsorption on the HS nanoclay, as well as the formation of insoluble metal hydroxide.

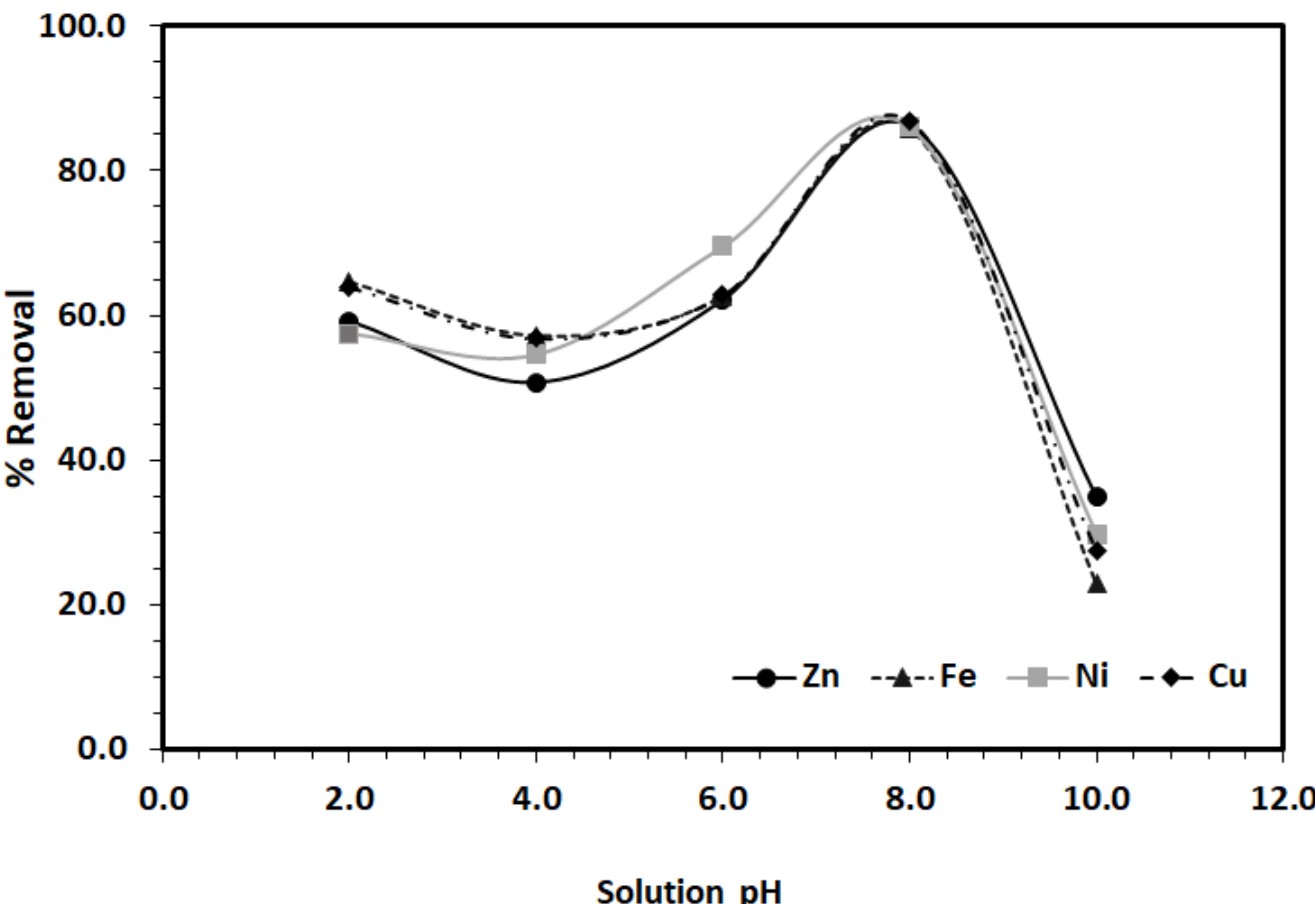

**Figure 8.** Effect of solution pH on the removal of metal ions (Zn, Fe, Ni, Cu) adsorption from a desalination plant outfall brine discharge by HS nanoclay. (Experimental conditions: 60.0 mL solution, 90 min adsorption time, 150 mg dosage, and 298 K).

The variation of the adsorption/removal percentage of Zn, Fe, Ni, Cu ions from the outfall brine discharge sample with solution temperature by HS nanoclay was investigated in a temperature range from 283 K to 323 K. The results showed that rising the solution temperature was associated with a significant increase in the removal till it reached its maximum at 323 K for all metal ions, as it is presented in Figure 9. This may indicate that the removal process is endothermic, which may be attributed to the fact that rising the solution temperature enhanced the diffusion of the ions from the bulk solution to the HS nanoclay surface, and enhanced the adsorption/removal process.

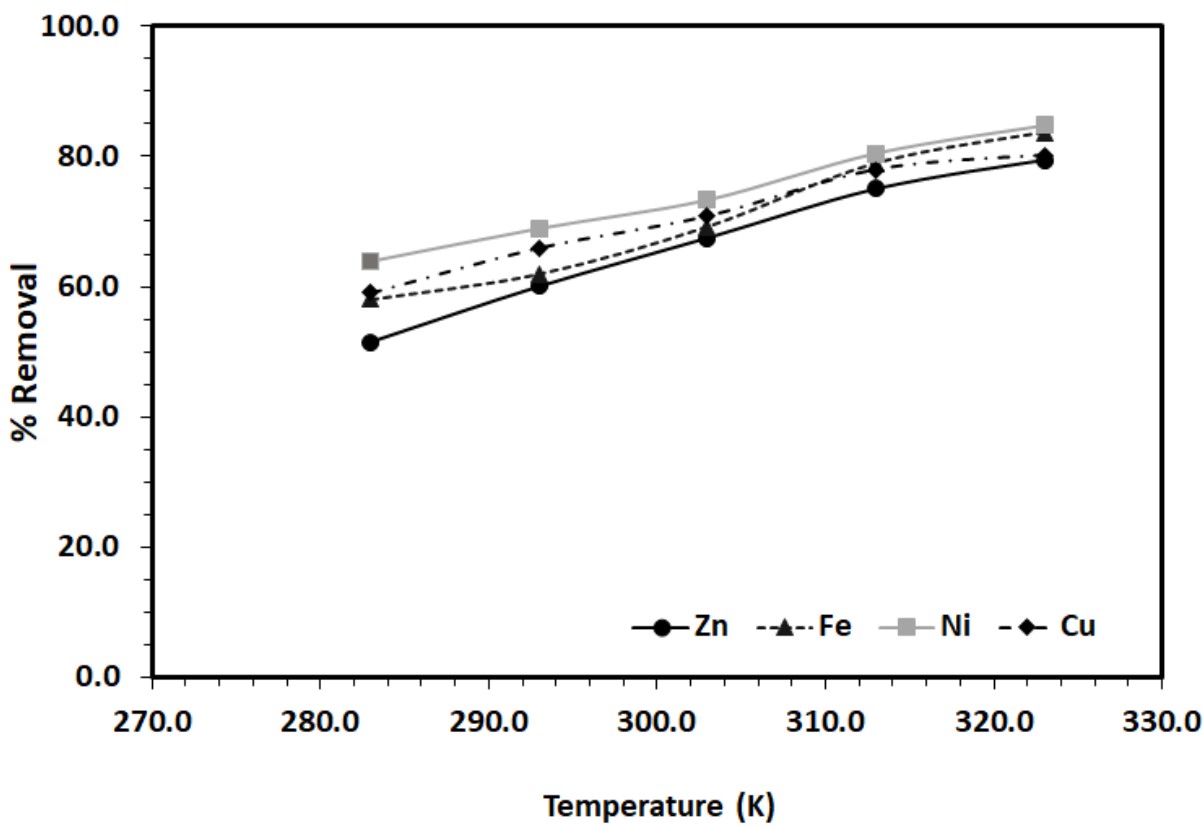

**Figure 9.** Effect of solution temperature on the removal of metal ions (Zn, Fe, Ni, Cu) adsorption from a desalination plant outfall brine discharge by HS nanoclay. (Experimental conditions: 60.0 mL solution, 90 min adsorption time, 150 mg dosage, and pH 8.0).

*3.4. Kinetics Study*

A kinetics study is very crucial for the understanding of the adsorption/removal of pollutants such as heavy metal ions; Zn, Fe, Ni, Cu ions, by a solid adsorbent such as HS nanoclay. To explore the process kinetically helps develop an appropriate mathematical model for a better description of the pollutant/solid adsorbent interactions in order to design the suitable adsorbent materials for environmental remediation. Figure 10 shows the variation of the experimental removal capacities, i.e., the amount of metal ions adsorbed from the desalination outfall sample by HS nanoclay ($q_t$) with interaction time. It was found that the adsorption/removal of Zn, Fe, Ni, Cu ions by HS nanoclay reached equilibrium within 90 min, and further extension of the contact time did not change the removal capacities significantly. The adsorption experimental data shown in Figure 10 were treated kinetically using the Lagergren pseudo-first-order (PFO) kinetic model [40], and the pseudo-second-order (PSO) kinetic model [41,42], as these are the most common and well-known kinetic models used in order to recognize the nature of the removal process.

The linearized forms of the PFO kinetic model (Equation (3)), and the pseudo-second-order (PSO) kinetic model (Equation (4)) are given as:

$$\ln(q_e - q_t) = \ln q_e - k_1 t \tag{3}$$

$$\frac{t}{q_t} = \frac{1}{k_2\, q_e^2} + \frac{t}{q_e} \tag{4}$$

where $k_1$ (min$^{-1}$) is the PFO adsorption rate coefficient, $k_2$ (g/(mg·min)) is the PSO rate coefficient, and $q_e$ and $q_t$ are the values of the amount of the metal ion removed per unit mass of HS nanoclay at equilibrium, at any time $t$, respectively. Applying Equation (3) to the experimental adsorption data in Figure 10, and plotting ln ($q_e - q_t$) vs. $t$ did not

converge well in most of the metal ions, and an unacceptable correlation coefficient was obtained, as it is presented in Table 1 and Figure 11. Meanwhile, applying the PSO kinetic model to the adsorption experimental data in Figure 10 and plotting $\frac{t}{q_t}$ vs. $t$ according to Equation (4), it converged very well, and for a straight line an excellent $R^2$ value was achieved, as it is presented in Table 1 and Figure 12.

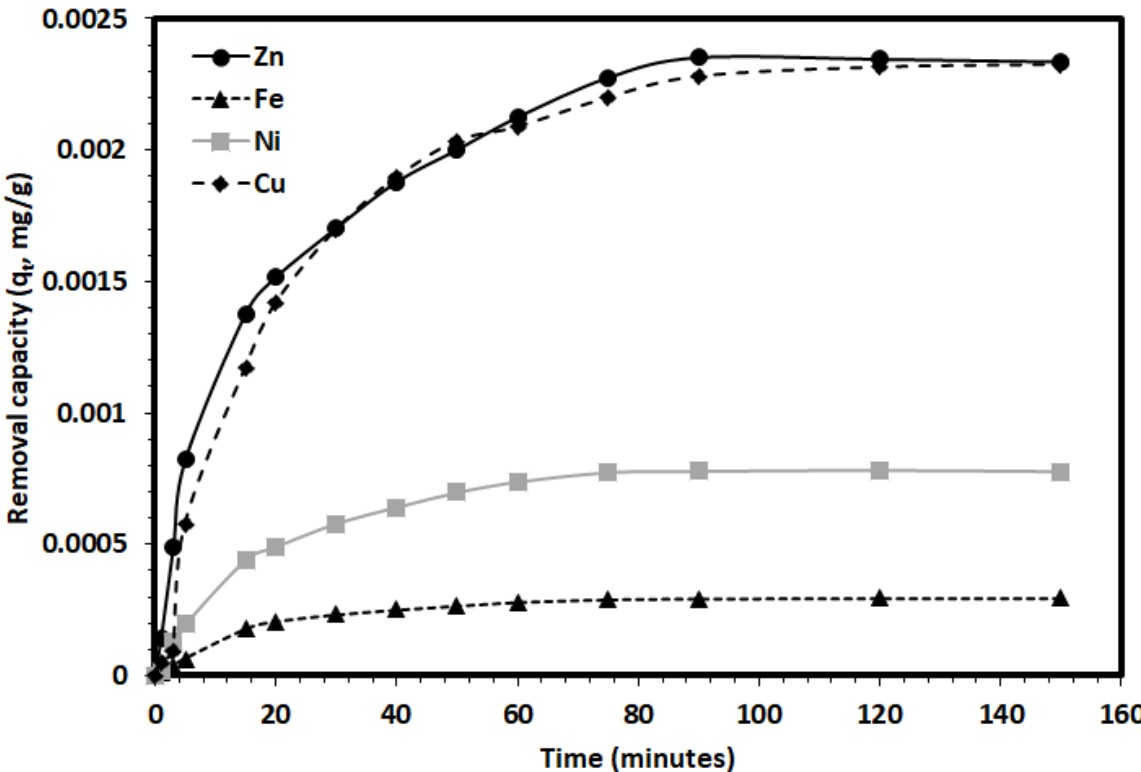

**Figure 10.** The variation of the removal capacities with time for the removal of Zn, Fe, Ni, Cu ions from a desalination plant outfall brine discharge by HS nanoclay. (Experimental conditions: 60.0 mL solution, pH 8.0, 150 mg dosage, and 298 K).

**Table 1.** Parameters of the pseudo-first-order (PFO), and pseudo-second-order (PSO) kinetic models for the removal of Zn, Fe, Ni, Cu ions from the outfall brine sample using HS nanoclay.

| Parameter | PFO Kinetic Model | | | | PSO Kinetic Model | | | |
|---|---|---|---|---|---|---|---|---|
| | Zn Ions | Fe Ions | Ni Ions | Cu Ions | Zn Ions | Fe Ions | Ni Ions | Cu Ions |
| $k_1$ | $3.7 \times 10^{-2}$ | $4.1 \times 10^{-2}$ | $3.3 \times 10^{-2}$ | $3.8 \times 10^{-2}$ | – | – | – | – |
| $k_2$ | – | – | – | – | 34.2 | 183 | 64.8 | 34.9 |
| $q_{e,exp}$ (mg/g) | $2.4 \times 10^{-3}$ | $3.0 \times 10^{-4}$ | $8.0 \times 10^{-4}$ | $2.3 \times 10^{-3}$ | $2.4 \times 10^{-3}$ | $3.0 \times 10^{-4}$ | $8.0 \times 10^{-4}$ | $2.3 \times 10^{-3}$ |
| $q_{e,calc}$ (mg/g) | $1.8 \times 10^{-3}$ | $2.0 \times 10^{-4}$ | $6.0 \times 10^{-4}$ | $2.1 \times 10^{-3}$ | $2.4 \times 10^{-3}$ | $3.0 \times 10^{-4}$ | $9.0 \times 10^{-4}$ | $2.5 \times 10^{-3}$ |
| $R^2$ | 0.874 | 0.971 | 0.895 | 0.993 | 0.992 | 0.974 | 0.963 | 0.991 |
| $\chi^2$ | $1.5 \times 10^{-4}$ | $1.5 \times 10^{-5}$ | $9.0 \times 10^{-5}$ | $1.7 \times 10^{-5}$ | $1.3 \times 10^{-5}$ | $5.1 \times 10^{-6}$ | $1.3 \times 10^{-5}$ | $1.2 \times 10^{-5}$ |
| SSE | $2.8 \times 10^{-7}$ | $3.6 \times 10^{-9}$ | $5.1 \times 10^{-8}$ | $3.7 \times 10^{-8}$ | $3.4 \times 10^{-8}$ | $1.7 \times 10^{-9}$ | $1.2 \times 10^{-8}$ | $3.1 \times 10^{-8}$ |

The appropriateness of the PSO kinetic model in comparison to the PFO kinetic model for the removal of Zn, Fe, Ni, Cu ions from the outfall brine sample using HS nanoclay was validated using two statistical tests, the chi-square test [43]; Equation (5), and the sum of the squares of errors (SSE) [44]; Equation (6).

$$\chi^2 = \sum \frac{\left(q_{e,calc} - q_{e,exp}\right)^2}{q_{e,calc}} \tag{5}$$

$$\text{SSE} = \sum_{i=1}^{n} \left(q_{e,calc}^i - q_{e,exp}^i\right)^2 \tag{6}$$

where $q_{e,calc}$ and $q_{e,exp}$ are the calculated and experimental amounts of removed Zn, Fe, Ni, Cu ions per unit mass of HS nanoclay at equilibrium. For the Zn ions removal, the $\chi^2$ values obtained were $1.5 \times 10^{-4}$ and $1.3 \times 10^{-5}$, and the SSE values were $2.8 \times 10^{-7}$ and $3.4 \times 10^{-8}$ for the PFO and the PSO, respectively, whereas for Fe ions removal, the $\chi^2$ values were $1.5 \times 10^{-5}$ and $5.1 \times 10^{-6}$, and the SSE values were $3.6 \times 10^{-4}$ and $1.7 \times 10^{-4}$ for the PFO and the PSO, respectively, for Ni ions removal, the $\chi^2$ values were $9.0 \times 10^{-4}$ and $1.3 \times 10^{-5}$, and the SSE values were $5.1 \times 10^{-8}$ and $1.2 \times 10^{-8}$ for the PFO and the PSO, respectively, and finally for Cu ions removal, the $\chi^2$ values were $1.7 \times 10^{-5}$ and $1.2 \times 10^{-5}$, and the SSE values were $3.7 \times 10^{-8}$ and $3.1 \times 10^{-8}$ for the PFO and the PSO, respectively, as it is shown in Table 1. These results, in addition to the excellent regression coefficients values of the PSO kinetic models compared with the PFO kinetic models, indicated the appropriateness of the PSO kinetic model for describing the removal of Zn, Fe, Ni, and Cu ions by HS nanoclay from the desalination plant outfall discharge brine sample. In this study, the better applicability of the pseudo-second-order model compared with the pseudo-first-order model for the description of the removal process was observed for halloysite nanoclay. In other studies, it was also used for the removal of oxytetracycline antibiotic [45], methylene blue dye [46,47], Orange G dye [48], lead (II) [49], phosphate [50], and nitrate ions [51]. This indicates that the removal process depends on both the heavy metal ions concentration as well as the number of active sites available on the HS nanoclay.

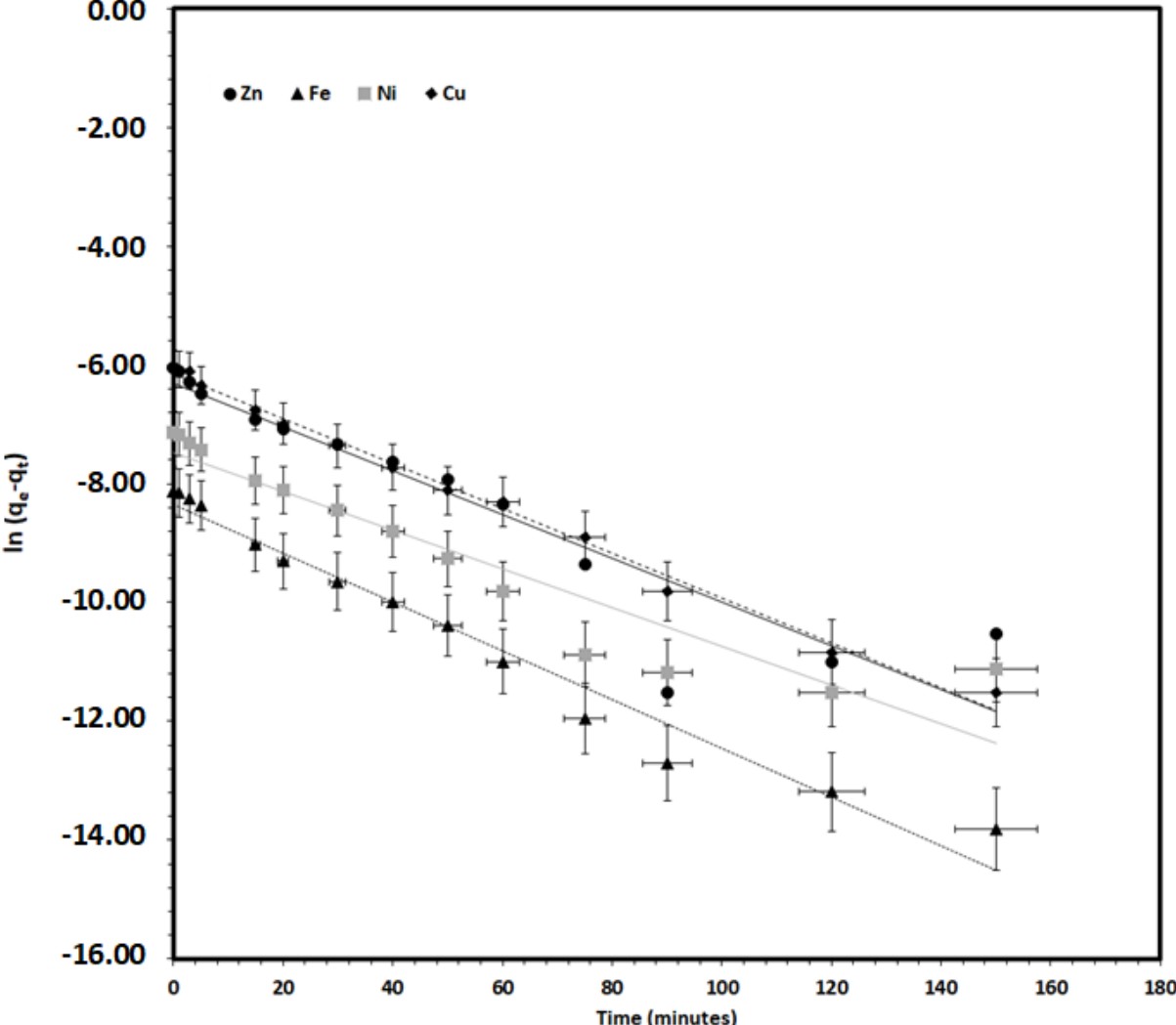

**Figure 11.** The applications of the PFO kinetic model for the removal of Zn, Fe, Ni, Cu ions from a desalination plant outfall brine discharge by HS nanoclay. (Experimental conditions: 60.0 mL solution, pH 8.0, 150 mg dosage, and 298 K).

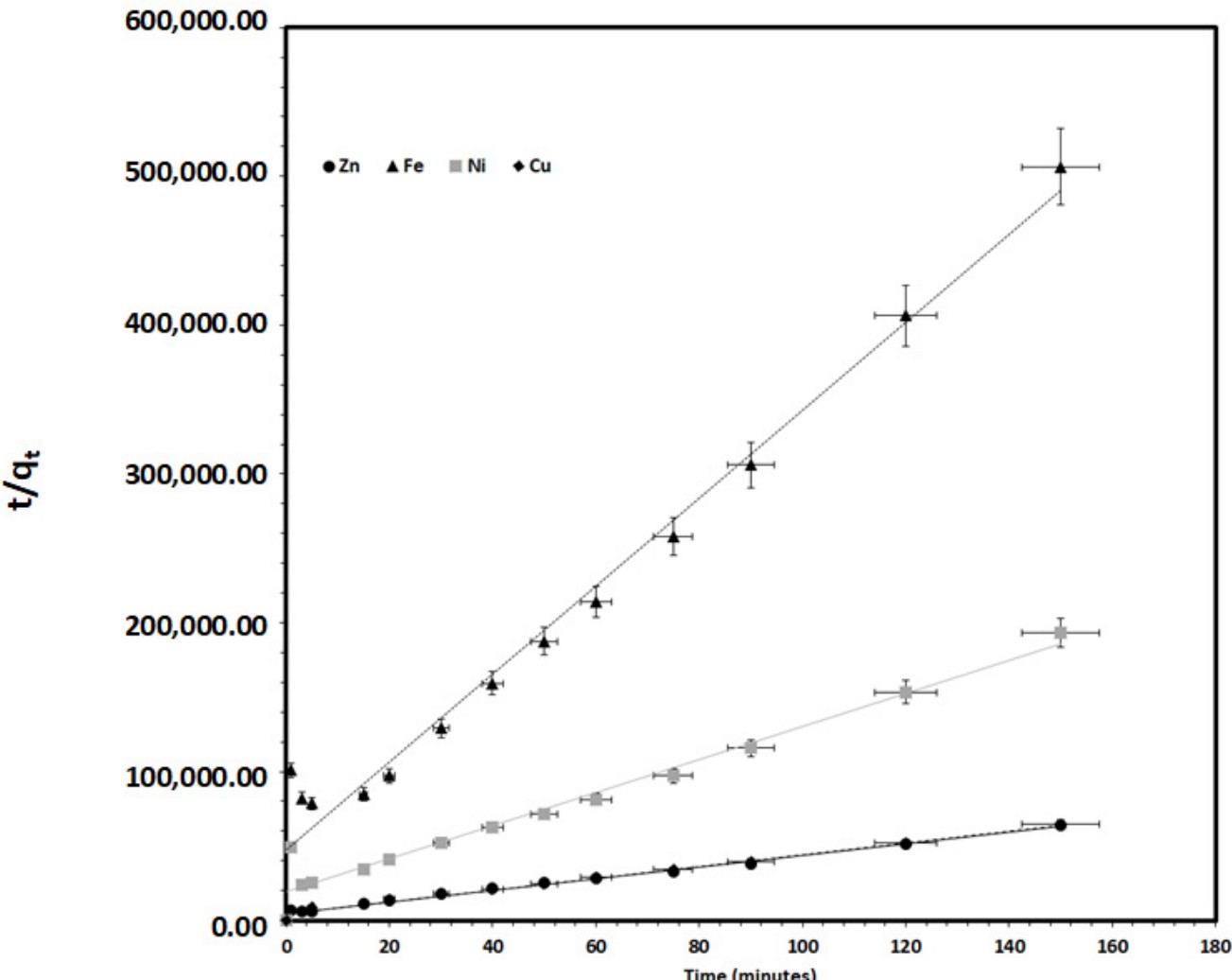

**Figure 12.** The applications of the PSO kinetic model for the removal of Zn, Fe, Ni, Cu ions from a desalination plant outfall brine discharge by HS nanoclay. (Experimental conditions: 60.0 mL solution, pH 8.0, 150 mg dosage, and 298 K).

Based on the above results, the optimum removal of Zn, Fe, Ni, and Cu ions by HS nanoclay from the desalination plant outfall discharge brine sample could be achieved using 150 mg of the HS nanoclay, within 90 min, and at ambient conditions of solution temperature and pH. In addition, as it is presented in Figure 13, the treatment of the desalination plant outfall brine discharge sample with HS nanoclay highly decreased the concentrations of the heavy metals, Zn, Fe, Ni, and Cu ions, as well as the salinity. The concentrations were lower than the ones obtained for the control sample. The Zn ion concentration was reduced from 103.9 nM for the outfall brine discharge sample to 12.76 nM after treatment, which is significantly lower than the concentration in the control sample; 45.2 nM, meanwhile for Fe ion, the concentration was reduced from 15.3 nM for the outfall brine discharge sample to 1.57 nM after treatment, which is much lower than the concentration in the control sample; 3.67 nM, whereas for Ni ion, the concentration was reduced from 39.4 nM for the outfall brine discharge sample to 4.91 nM after treatment, which is greatly lower than the concentration in the control sample; 13.9 nM, and for Cu ion, the concentration was reduced from 117.1 nM for the outfall brine discharge sample to 13.1 nM after treatment, which is much lower than the concentration in the control sample; 28.6 nM. In addition, the salinity was reduced from 48.3 ppt for the outfall brine discharge sample to 41.2 ppt after treatment, which is lower than the concentration in the control sample; 41.4 ppt. This may indicate that the possible application of HS nanoclay for environmental remediation of the desalination plant outfall brine discharge of both heavy

metals and salinity is highly efficient. In addition, in comparison with other technologies used for the removal of heavy metals from the brine discharge, which are based on the application of membranes [25,26], the current method which is based on adsorption on HS nanoclay can be easily applied and has the ability to regenerate and reuse both metal ions and the HS nanoclay. Moreover, regarding the removal mechanism, and based on the current results, one could propose that the removal mechanism of the Zn, Fe, Ni, and Cu ions from the brine discharge using the HS nanoclay is mainly based on the electrostatic attraction forces between the positively charged metal ions; $Zn^{2+}$, $Fe^{2+}$, $Ni^{2+}$ and $Cu^{2+}$, and the silanol groups (Si–O–H) present at the HS nanoclay surface [48].

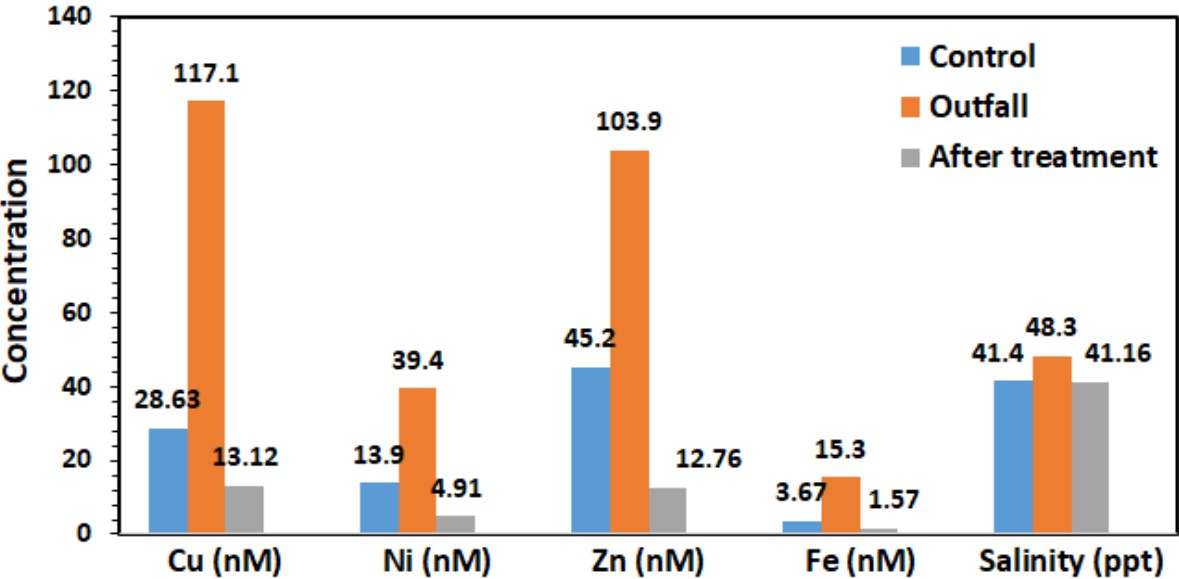

**Figure 13.** Comparison of metal ions and salinity concentrations of the desalination plant outfall brine discharge with the treatment by HS nanoclay before and after treatment. (Experimental conditions: 60.0 mL solution, 90 min adsorption time, 150 mg dosage, and pH 8.0, at 25 °C temperature).

## 4. Conclusions

The removal efficiency of heavy metals and salinity were successfully established by the use of Halloysite nanoclay. The characterization techniques showed the nanostructure of the HS nanoclay with hollow tubes and an average diameter of 60 nm and an average length of 10 microns, the existence of XRD characteristic peaks, as well as a high specific surface of 72.8 $m^2$ $g^{-1}$. HS nanoclay was used for the treatment of the desalination plant outfall brine discharge of both heavy metals and salinity, and the removal operation parameters were optimized. It was found that most of the heavy metals and salinity were removed from the desalination plant outfall brine discharge and the final concentrations were lower than the concentrations in the control sample. This was achieved by using 150 mg of HS nanoclay within 90 min, and at ambient conditions. In addition, the removal process was explored kinetically, and it was found that the pseudo-second-order model is more suitable for the description of the removal process compared with the pseudo-first-order model. Finally, the present work established the ability of the nanoclay to treat and remove the discharge's salinity as well as the heavy metals to be lower than the standard sample.

**Author Contributions:** N.S.A., Data curation, software, methodology, writing; R.K.A.-F., Writing—reviewing & editing; I.I.S., Reviewing & editing; B.A.A.-M., Validation, formal analysis; Y.N.K., Investigation, software, methodology, writing; M.A.S., Supervision, funding acquisition, conceptualization, writing—reviewing & editing. All authors have read and agreed to the published version of the manuscript.

**Funding:** This research work was funded by the Institutional Fund Projects under grant no. (IFPHI-181-130-2020). Therefore, the authors gratefully acknowledge the technical and financial support from the Ministry of Education and King Abdulaziz University, DSR, Jeddah, Saudi Arabia.

**Institutional Review Board Statement:** Not applicable.

**Informed Consent Statement:** Not applicable.

**Data Availability Statement:** The data presented in this study is available on request from the corresponding author.

**Acknowledgments:** This research work was funded by the Institutional Fund Projects under grant no. (IFPHI-181-130-2020). Therefore, the authors gratefully acknowledge the technical and financial support from the Ministry of Education and King Abdulaziz University, DSR, Jeddah, Saudi Arabia.

**Conflicts of Interest:** The authors declare no conflict of interest.

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
