# Peer review of "Environmental Remediation of Desalination Plant Outfall Brine Discharge from Heavy Metals and Salinity Using Halloysite Nanoclay"

_water, doi:10.3390/w13070969_

Round 1

Reviewer 1 Report

The article is well constructed and written. The introduction and the research part are well documented. The research methods and test results are clearly described, supported by proper conclusion and reference to the results of other authors. I recommend the publication of the work  in the current form with minor technical corrections (comments bellow) .

Line 155: Figure 5 shows other results.

Line 167: these results are shown in Figure 5.

Line 180: these results are shown in Figure 6.

Figure 10: reference in the text; what about Cu ions?

Line 195: develop a conclusion, to what standard?

Author Response

Reviewer 1:

Comments and Suggestions for Authors

The article is well constructed and written. The introduction and the research part are well documented. The research methods and test results are clearly described, supported by proper conclusion and reference to the results of other authors. I recommend the publication of the work in the current form with minor technical corrections (comments bellow).

Line 155: Figure 5 shows other results.

Authors:

The sentence was corrected, per the prominent reviewer’s advice.

Line 167: these results are shown in Figure 5.

Authors:

A full revision of the manuscript was performed, and the sentence was corrected, per the prominent reviewer’s advice.

Line 180: these results are shown in Figure 6.

Authors:

A full revision of the manuscript was performed, and the sentence was corrected, per the prominent reviewer’s advice.

Figure 10: reference in the text; what about Cu ions?

Authors:

Cu and Zn ions almost had the same trend, therefore, the authors changed the figure symbols to contain both Cu and Zn ions clearly.

Line 195: develop a conclusion, to what standard?

Authors:

According to the results, rising the solution temperature associate with a significant increase in the removal till it reached its maximum at 323 K for all metal ions

Reviewer 2 Report

Dear Authors,

 In my opinion yourwork  is a great effort worthy of publication at this journal, nevertheless after major revision  since it is mispresented.  More details are listed below along with other specific comments:

Introduction: Need reorganization

Section 2.1: Please provide a map if it is possible

Sections 2.1 &2.2 sections could be merged in one named “Materials”

Lines 95-98: Specify the parameters (amount, time, temperature)

Line 112: Please explain the abbreviation PTFE, what is this stands for?

Section 3.1: Kaolinite group minerals were not discriminated easy by XRD, if you have its respect IR spectra please provide it.

Sections 3.2, 3.3: Please introduce tables as analytical as possible. The amount of numbers in text is quite hard to follow.

Line 153, 158: Choose to use symbol for chemical elements or words in the text, please check

Line 167: Fig 6 change with Fig 5

Line 180: Fig 7 change with Fig 6

 In my opinion your discussion is too poor compared of your interesting results. I think you could enrich your discussion on the mechanism of adsorption in accordance with the studied parameters or you can discuss on the competitiveness of the studied chemical elements and the adsorption.

I am really looking forward to see this interesting article published.

Kind regards

Author Response

Reviewer 2:

Comments and Suggestions for Authors

Dear Authors,

 In my opinion yourwork  is a great effort worthy of publication at this journal, nevertheless after major revision  since it is mispresented.  More details are listed below along with other specific comments:

Introduction: Need reorganization

Authors:

The introduction was organized as the following:

  • Explain the problem of Freshwater resource scarcity
  • The necessity of desalination plant
  • Problems associated with desalination plant
  • Major technologies for brine discharge treatment
  • The lack of information regarding the removal of heavy metals from brine discharge
  • The objective and novelty of the current work

Section 2.1: Please provide a map if it is possible

Authors:

A map was provided to the collection site, as Figure 1.

Sections 2.1 &2.2 sections could be merged in one named “Materials”

Authors:

The two sections were merged in one named “Materials” section.

Lines 95-98: Specify the parameters (amount, time, temperature)

Authors:

The parameters are different for each experiments, therefore, the authors specified the parameters at each figure caption as the Experimental conditions. For example at Figure 5, Experimental conditions: 60.0 ml solution, pH 8.0, 90 min adsorption time, and 25 °C temperature.

Line 112: Please explain the abbreviation PTFE, what is this stands for?

Authors:

The abbreviation PTFE was explained as Polytetrafluoroethylene at the manuscript.

Section 3.1: Kaolinite group minerals were not discriminated easy by XRD, if you have its respect IR spectra please provide it.

Authors:

The abbreviation PTFE was explained as Polytetrafluoroethylene at the manuscript.

Sections 3.2, 3.3: Please introduce tables as analytical as possible. The amount of numbers in text is quite hard to follow.

Authors:

Table 1 was revised and the required changed

Line 153, 158: Choose to use symbol for chemical elements or words in the text, please check

Authors:

The required changes were performed, and the symbols for chemical were used, per the prominent reviewer comment.

Line 167: Fig 6 change with Fig 5

Authors:

A full revision of the manuscript was performed, and the sentence was corrected, per the prominent reviewer’s advice.

Line 180: Fig 7 change with Fig 6

Authors:

A full revision of the manuscript was performed, and the sentence was corrected, per the prominent reviewer’s advice.

 In my opinion your discussion is too poor compared of your interesting results. I think you could enrich your discussion on the mechanism of adsorption in accordance with the studied parameters or you can discuss on the competitiveness of the studied chemical elements and the adsorption.

Authors:

The discussion was revised, and the following paragraphs were added to the manuscript:

Based on the literature, the applicability of the pseudo-second-order model compared with the pseudo-first-order model for the description of the removal process was observed for the removal of hallosyte nanoclay was used for the removal oxytetracycline antibiotic [44], methylene blue dye [45,46], Orange G dye [47], lead(II) [48], phosphate [49], and nitrate ions [50], indicating that the removal process depends on both the heavy metal ions concentration, as well as the number of active sites available on the HS nanoclay.

Also, in comparison with other technologies used for the removal of heavy metals from the brine discharge, which based on the application of membrane [25,26], the current method which based on adsorption on HS nanoclay characterized with the ease of application, as well as the ability to regenerate, and reuse both metal ions and the HS nanoclay. Moreover, regarding the removal mechanism, and based on the current results, it could propose that the removal mechanism of the Zn, Fe, Ni, and Cu ions from the brine discharge using the HS nanoclay mainly based on the electrostatic attraction forces between the positively charged metal ions; Zn2+, Fe2+, Ni2+ and Cu2+, and the silanol groups (Si–O–H) presents at the HS nanoclay surface [48].

Reviewer 3 Report

In this manuscript, the authors reported the Environmental remediation of desalination plant outfall brine discharge from heavy metals and salinity using Halloysite nanoclay. The paper seems good, however, some points need to be clarified in the manuscript before being recommended to be published in the journal. Also, details are needed on their discussion. Q1. What is the novelty of this work explain it? Q2. The authors should mention the concentration of metal ions which is used in this work. Q3. The author confirmed that pseudo-second-order model is suitable for the description of the removal process compared with the pseudo-first-order model. Explain with more details and add some references? Q4. I find that all figures 9 and 10 contain data without error bars, it is better to include error bars in the figures. Q5. Compare your work with other reported work and explain the advantages using Halloysite (HS) nanoclay. Q6. In Figure.11 why Cu and Zn concentration is very low when compared with before treatment. Explain it?

Author Response

Comments and Suggestions for Authors

In this manuscript, the authors reported the Environmental remediation of desalination plant outfall brine discharge from heavy metals and salinity using Halloysite nanoclay. The paper seems good, however, some points need to be clarified in the manuscript before being recommended to be published in the journal. Also, details are needed on their discussion.

Q1. What is the novelty of this work explain it?

Authors:

The novelty of the work was highlighted at the objective part of the introduction, as the current work used for the first time Halloysite nanoclay for the treatment of the outfall brine discharge sample of Yanbu Desalination Plant (YDP), Saudi Arabia, from both heavy metals; zinc, iron, nickel, and copper, as well and salinity to the permissible levels. As the following:

“The present research work objective is to explore, for the first time, the potential application of Halloysite nanoclay (HS nanoclay) for the treatment of the outfall brine discharge sample of Yanbu Desalination Plant (YDP), Saudi Arabia, from both heavy metals; zinc, iron, nickel, and copper, as well and salinity to the permissible levels.”

Q2. The authors should mention the concentration of metal ions which is used in this work.

Authors:

The authors used the real outfall brine discharge sample of Yanbu Desalination Plant (YDP), Saudi Arabia, and the concentrations of the zinc, iron, nickel, and copper, as well and salinity were measured before treatment and explained at section 3.2. Outfall brine discharge and control samples characterization as the following:

The concentrations of the heavy metals of the brine discharge sample were measured and were found to be 103.8 nM (6.8 ng/mL, ppb), 15.3 nM (0.85 ng/mL), 39.4 nM (2.3 ng/mL), and 117.1 nM (7.4 ng/mL), for the zinc, iron, nickel, and copper ions, respectively, whereas the salinity concentration was 48.0 g/L (ppt). On the other hand, the concentrations of the heavy metals of the the control sample were 45.2 nM (3.0 ng/mL), 3.67 nM (0.20 ng/mL), 13.9 nM (0.81 ng/mL), and 28.6 nM (1.8 ng/mL), for the zinc, iron, nickel, and copper ions, respectively, with salinity concentration of 42.2 g/L (ppt).

Q3. The author confirmed that pseudo-second-order model is suitable for the description of the removal process compared with the pseudo-first-order model. Explain with more details and add some references?

Authors:

The following paragraph and references were added to the manuscript, per the prominent reviewer recommendation:

Based on the literature, the applicability of the pseudo-second-order model compared with the pseudo-first-order model for the description of the removal process was observed for the removal of hallosyte nanoclay was used for the removal oxytetracycline antibiotic [44], methylene blue dye [45,46], Orange G dye [47], lead(II) [48], phosphate [49], and nitrate ions [50], indicating that the removal process depends on both the heavy metal ions concentration, as well as the number of active sites available on the HS nanoclay.

[44] S. Ramanayaka, B. Sarkar, A. T. Cooray, Y. S. Oke, M. Vithanage Halloysite nanoclay supported adsorptive removal of oxytetracycline antibiotic from aqueous media,  Journal of Hazardous Materials 384 (2020) 121301.

[45] S. Radoor, J. Karayil, J. Parameswaranpillai, S. Siengchin, Adsorption of methylene blue dye from aqueous solution by a novel PVA/CMC/halloysite nanoclay bio composite: Characterization, kinetics, isotherm and antibacterial properties. J Environ Health Sci Engineer 18 (2020) 1311–1327.

[46] T. Ngulube, J. R. Gumbo, V. Masindi, A. Maity, Preparation and characterisation of high performing magnesite-halloysite nanocomposite and its application in the removal of methylene blue dye, Journal of Molecular Structure 1184 (2019) 389-399.

[47] M. Abdel Salam, S. Kosa, A. Al-Beladi, Application of nanoclay for the adsorptive removal of Orange G dye from aqueous solution, Journal of Molecular Liquids 241 (2017) 469-477.

[48] S. Cataldo, G. Lazzara, M. Massaro, N. Muratore, A. Pettignano, S. Riela, Functionalized halloysite nanotubes for enhanced removal of lead(II) ions from aqueous solutions, Applied Clay Science Volume 156 (2018) 87-95.

[49] D. A. Almasri, N. B. Saleh, M. A. Atieh, G. McKay, S. Ahzi, Adsorption of phosphate on iron oxide doped halloysite nanotubes. Scientific Reports 9 (2019) 3232.

[50] H. Parab, K. Chauhan, J. Ramkumar, R. Devi P.S., N. S. Shenoy, S. D. Kumar,  In-situ synthesised polyaniline - halloysite nanoclay composite sorbent for effective decontamination of nitrate from aqueous streams International Journal of Environmental Analytical Chemistry https://doi.org/10.1080/03067319.2020.1828390

Q4. I find that all figures 9 and 10 contain data without error bars, it is better to include error bars in the figures.

Authors:

Error bars were added to figures 9 and 10, per the prominent reviewer’s recommendation.

Q5. Compare your work with other reported work and explain the advantages using Halloysite (HS) nanoclay.

Authors:

The following paragraph was added to the discussion part:

Also, in comparison with other technologies used for the removal of heavy metals from the brine discharge [25,26] which based on the application of membrane [25], the current method which based on adsorption on HS nanoclay characterized with the ease of application, as well as the ability to regenerate, and reuse both metal ions and the HS nanoclay.

Q6. In Figure.11 why Cu and Zn concentration is very low when compared with before treatment. Explain it?

Authors:

For all the heavy metal ions, the treatment of the brine discharge greatly decreased their concentration to be lower than the control itself, and this not only limited to Cu and Zn, but to Ni, and Fe ions as well as it was well explained at the following section.

Also, as it is presented in Figure 12, the treatment of the desalination plant outfall brine discharge sample with HS nanoclay highly decreased the concentrations of the heavy metals, Zn, Fe, Ni, and Cu ions, as well as the salinity to lower than the control sample. The Zn ion concentration was reduced from 103.9 nM for the outfall brine discharge sample to 12.76 nM after treatment, which is significantly lower than the control sample; 45.2 nM, meanwhile for Fe ion, the concentration was reduced from 15.3 nM for the outfall brine discharge sample to 1.57 nM after treatment, which is much lower than the control sample; 3.67 nM, whereas for Ni ion, the concentration was reduced from 39.4 nM for the outfall brine discharge sample to 4.91 nM after treatment, which is greatly lower than the control sample; 13.9 nM, and for Cu ion, the concentration was reduced from 117.1 nM for the outfall brine discharge sample to 13.1 nM after treatment, which is much lower than the control sample; 28.6 nM.

Round 2

Reviewer 2 Report

 Dear authors,

the  manuscript has been revised quite statisfied. In my opinion this version can be published in present form.

Kind regards 

Reviewer 3 Report

Accepted